# Looking beyond Pertussis in Prolonged Cough Illness of Young Children

**DOI:** 10.3390/vaccines10081191

**Published:** 2022-07-27

**Authors:** Rajlakshmi Viswanathan, Sanjay Bafna, Manohar Lal Choudhary, Monika Reddy, Savita Katendra, Shradha Maheshwari, Sheetal Jadhav

**Affiliations:** 1Bacteriology Group, ICMR-National Institute of Virology, 130/1, Sus Road, Pashan, Pune 411021, India; sairajsavita@gmail.com (S.K.); shradhamaheshwari1992@gmail.com (S.M.); 2Department of Pediatric Medicine, Jehangir Hospital, 32, Sassoon Road, Pune 411001, India; sanjaybafna16@gmail.com (S.B.); monika.reddy39@gmail.com (M.R.); 3Influenza Group, ICMR-National Institute of Virology, 20-A, Ambedkar Road, Pune 411001, India; mlchoudhary@gmail.com (M.L.C.); sheetalk86@gmail.com (S.J.)

**Keywords:** whooping cough, *Bordetella pertussis*, *Bordetella holmesii*, respiratory syncytial virus, infant, cough illness

## Abstract

Pertussis, commonly known as whooping cough, is one of the most poorly controlled vaccine-preventable diseases in the world. South-East Asia is estimated to contribute the most to childhood disease burden while this remains largely unexplored in India. The clinical diagnosis of pertussis in young children is a challenge as the classical four-stage presentation with paroxysmal cough or whoop may be absent. It is also difficult to differentiate from other respiratory infections which can cause pertussis-like illness. Children below two years with prolonged cough illness attending an urban pediatric center in western India, were evaluated for pertussis and viral infections by molecular methods. *Bordetella pertussis* and *B. holmesii* were confirmed in three each of 45 suspected cases, and RSV-A and hMPV were the most common viruses that were detected. These organisms can mimic mild cases of pertussis and need to be considered in differential diagnosis of prolonged cough illness in young children. The accurate etiology of prolonged cough illness needs to be detected and documented to ensure appropriate management and accurate estimates of disease burden.

## 1. Introduction

Pertussis is a major public health problem with South-East Asia estimated to contribute the most to its global childhood disease burden [1]. There is paucity of information on pertussis and infection by other species of *Bordetella* from India [2]. The whole cellular pertussis (wP) vaccine was introduced in India under the expanded program of immunization (EPI) in 1978. The primary immunization with the wP vaccine at 6, 10, and 14 weeks of age is followed by boosters at 16–24 months and 4–5 years of age [3]. There is no national policy for booster immunization of pregnant women or any specific age group. The limited usage of the acellular pertussis vaccine (aP) booster at 10 years and during pregnancy exists in the private sector. The estimated coverage is more than 80% for DPT3 immunization in 67% of all the districts of the country [4]. Although no systematic information on pertussis is available from India, it is estimated that India alone has contributed 37,274 pertussis cases to the global burden in 2016 [5]. Despite this, very little information on pertussis is available from India in recent years [2,6]. The diagnosis of pertussis largely remains clinical, as laboratory services are usually not available except in a few apex institutions. A recent multicenter hospital-based surveillance has reported 4.62% laboratory-confirmed pertussis among the recruited infants [7].

Considering that the cost and complexity of laboratory confirmation remains a deterrent for timely diagnosis, we worked on the establishment of pertussis laboratory diagnostics in a phased manner starting with one site in India. The occurrence of pertussis-like illness of varied etiology has been reported [8], emphasizing the need to look at other infectious causes of prolonged cough illness. We report such a study in young children less than two years of age, attending an urban pediatric center in western India, focusing on *Bordetella pertussis*, other species of *Bordetella* and a panel of respiratory viruses.

## 2. Materials and Methods

A prospective hospital-based study was carried out at an urban pediatric center in Pune, Maharashtra, western India (Figure 1) with an outpatient attendance of 400–500 children per year. Young children below two years of age attending the outpatient department (OPD) or admitted were eligible for recruitment. The inclusion criteria were afebrile or minimally febrile cough illness lasting >14 days, with at least one of the following signs or symptoms: paroxysms of coughing, OR whooping, OR post-tussive vomiting, OR Apnea (with or without cyanosis for infants below one year of age) [9], OR cough illness of any duration that is clinically suspected to be pertussis by the physician. Written informed consent was obtained from parents for the data and sample collection. Clinical information, immunization history, and management were noted on a pre-designed form using Epi Info v7.2. A total of two nasopharyngeal (NP) specimens were collected from each participant using sterile flexible nylon flocked swabs (HiMedia Laboratories, Mumbai, India) and transported in cold chain within 24 h to the testing laboratory at the Indian Council of Medical Research-National Institute of Virology. One swab was in a sterile screw capped tube without any transport media for detection of *Bordetella pertussis* and other species of *Bordetella*. The second was in viral transport medium (HiMedia Laboratories, Mumbai, India) for detection of viral etiology.

The tip of the NP swab was eluted overnight in 600 µL phosphate buffered saline at room temperature (~22 °C), and 400 µL was used for DNA extraction by the Qiagen DNA mini kit (Qiagen, Germantown, MD, USA) [10]. Qualitative detection and differentiation for *B. pertussis, B. parapertussis,* and *B. holmesii* was performed by multiplex real-time PCR using oligonucleotide primers and conditions as described earlier [11]. Briefly, the multitarget real-time PCR assay, using *IS481* for *B. pertussis, B. parapertussis IS1001 (pIS1001)*, and *B. holmesii IS1001-like (hIS1001)* targets was performed in a single tube and interpreted as per the following algorithm (Table 1). *B. pertussis* was confirmed by testing for *ptxS1* [11]. To assess the quality of the sample, *RNaseP* was included as a target, using primers and probes that were described earlier [11]. 

Adapted with permission from Tatti et al. [11] 2011, American Society for Microbiology

Real-time reverse-transcription-polymerase chain reaction (qRT-PCR) assay was performed for influenza A [A(H1N1)pdm09 and A(H3N2)], influenza B, along with the housekeeping RNaseP gene [12], parainfluenza [PIV] virus 1, 2, 3, 4, human metapneumovirus (hMPV), respiratory syncytial virus (RSV) A&B, adenovirus, and rhinovirus [13]. RNA was extracted using MagMax-96 Viral RNA isolation kit (Thermo Fisher Scientific, Vilnius Lithuania). Nucleic acid amplification was performed by one-step qRT-PCR kit (SuperScript™ III kit, Invitrogen, Waltham, MA, USA) according to manufacturer’s protocol. A 25 μL PCR reaction mixture with 10 μmol each of forward and reverse primers, 5 μmol of TaqMan probe, 12.5 μL 2x buffer, 0.5 μL SuperScript™ III enzyme, and 5 μL nucleic acid templates was prepared. The thermal cycling conditions were 50 °C for 15 min for reverse transcription, initial denaturation at 94 °C for five minutes, 45 cycles of 3 steps, 15 s at 94 °C, 15 s at 50 °C, and 30 s at 55 °C incubation step during which fluorescence data were collected.

## 3. Results

Etiology: During the study period (December 2019 to January 2021), 51 participants were eligible for inclusion, of whom four did not consent. The sample collection was inadequate for two. Therefore, 45 participants were included for final analysis. A total of six of the 45 cases were confirmed as infected by *Bordetella* genus-three *Bordetella pertussis* and three *B. holmesii*. Co-infection with human metapneumovirus (hMPV) was observed in two children confirmed with *B. holmesii*, one of whom was also positive for Rhinovirus. Of the three cases of pertussis, two occurred in a pair of 15-month-old twins. One or more virus was detected in 17 cases. The most commonly detected was RSV-A (6/45) and hMPV (6/45).Clinical presentation: The age and gender of children that were detected with virus or *Bordetella* spp. was similar. All the children were well-nourished, except for two with viral infection who had moderate and severe acute malnutrition, respectively. There was no significant difference in the duration of the cough, paroxysmal and nocturnal cough, and post-tussive vomiting. Productive cough was significantly higher in children with viral infection (*p* = 0.0180) (Table 2). Complications such as earache, seizures, and skin rash did not occur. Out of six cases with *Bordetella* infection, two infants each with pertussis and *B. holmesii* infection had received age-appropriate vaccination for pertussis. The first case of pertussis was the child of migrant workers without any immunization records. None of these six cases reported any contact having prolonged cough illness (Table 3).Outcome: The majority of the cases presented in the OPD and one child with viral infection required admission. All of them were managed symptomatically. Azithromycin was used for treating the pertussis and *B. holmesii* cases. The six children recovered uneventfully. There was no mortality.

## 4. Discussion

Pertussis may not present as the characteristic whooping cough [14], rendering clinical diagnosis difficult at times. Laboratory confirmation is complicated since the gold standard of culture, albeit with high specificity but poor sensitivity, as well as the CDC-approved multiplex real-time PCR assay [11] are usually not available in laboratories across India. Recent studies have shown the occurrence of pertussis-like illness (PLI) [8] with varied etiology.

The present report is a study of infectious etiology of prolonged cough illness in children less than two years of age, which focused on the detection of *Bordetella* sp. along with viral etiology in children less than 2 years of age. We demonstrated the occurrence of pertussis in three of 45 suspected cases and *B. holmesii* infection in another three cases. Our data correlates with the findings of a recent multi-centre study from India, in which 4.62% suspected cases of pertussis were laboratory confirmed [7]. The comparatively lower positivity in our study could be due to differences in case definitions that were adopted in other hospital-based studies [7].

The suspected cases in our study were mildly symptomatic with only one child requiring admission; emphasizing the need to screen for pertussis in children presenting with mild but persistent cough, since such children can continue to be a source of infection for family members and contacts. Although immunization coverage for primary pertussis vaccination is high in India [3], the coverage is considerably lower for the booster doses. The role of pertussis vaccine in preventing infection with other species of *Bordetella* remains questionable and could explain the occurrence of *B. holmesii* infection in two patients with age-appropriate vaccination.

RSV-A and hMPV were the most commonly detected viral etiology in our study. These viruses have been earlier implicated in pertussis like illness [8]. Most of the RSV infections occurred between September to December, which is the usual season for respiratory viral infections in India. The clinical presentation of these cases was similar to the six cases of *Bordetella* infection, except for nature of the cough, emphasizing the need for laboratory confirmation of differential diagnosis (Table 2).

Our study had some limitations. As most of the cases were outpatients, leucocyte counts, which could provide a clue to pertussis diagnosis, could not be performed. Only a small number of cases were studied. However, this is the initial study from one of several centres and the data has been reported to provide a baseline information on pertussis and viral infections with a similar clinical profile. The expansion to three other sites across India is expected to generate comprehensive information.

In conclusion, our study reports baseline information from India on pertussis and pertussis-like illness that is caused by *B. holmesii* and several viruses. The indistinguishable clinical picture emphasizes the need for strengthening laboratory capacity for pertussis diagnosis as well as screening for at least RSV-A and hMPV in infants with prolonged cough illness.

## Figures and Tables

**Figure 1 vaccines-10-01191-f001:**
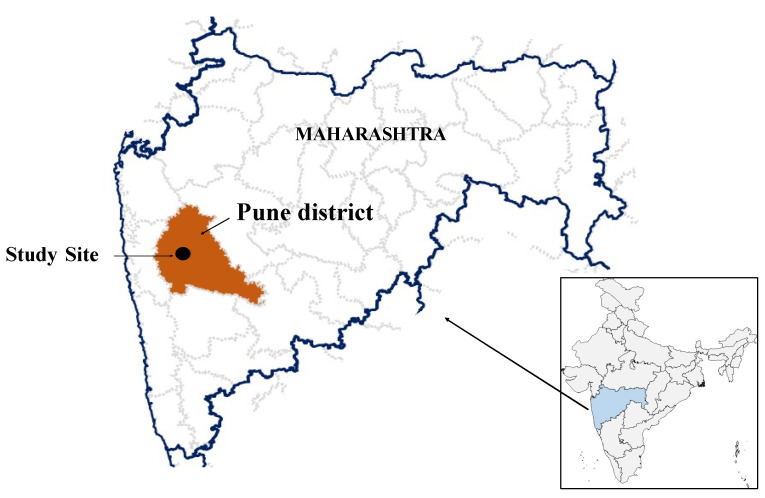
Location of Study Site.

**Table 1 vaccines-10-01191-t001:** Diagnostic algorithm for the detection and differentiation of *Bordetella* spp. [11].

Species	Multitarget Assay			Singleplex Assay *(ptxS1)*
	*IS481*	*pIS1001*	*hIS1001*	
*B. pertussis*	+	−	−	+
*B. parapertussis*	−	+	−	+
*B. holmesii*	+	−	+	−
*B. pertussis & B.parapertussis*	+	+	−	+
*B. pertussis & B.holmesii*	+	−	+	+

**Table 2 vaccines-10-01191-t002:** Comparison of demographic and the clinical features of the study population.

Variable	Infectious Etiology not Detected (*n* = 22)	Positive for *Bordetella (n* = 6)	Positive for Virus (*n* = 17)	*p*-Value
**Age (years)**				
≤1	8	2	7	0.9265
>1	14	4	10	
**Sex**				
Male	13	4	13	0.6294
Female	9	2	4	
**Duration of Cough (days)**				
14–21	12	5	16	0.5211
>21	10	1	1	
**Nature of cough**				
Dry	18	4	7	**0.01801 ***
Productive	4	2	10	
**Whoop**				
Yes	5	1	3	0.9036
No	17	5	14	
**Paroxysmal Cough**				
Yes	10	5	14	0.9566
No	12	1	3	
**Nocturnal cough**				
Yes	8	3	2	0.1149
No	14	3	15	
**Post-Tussive Vomiting**				
Yes	10	4	7	0.5535
No	12	2	10	

* Significant *p* value.

**Table 3 vaccines-10-01191-t003:** Profile of cough illness cases with infectious etiology.

Case	Age (m)/Sex	Nutrition	Duration of Cough (Weeks)	Type of Cough	Low Grade Fever	Age-Appropriate Vaccination for Pertussis	Any Contact with Cough Illness	*Bordetella* spp. Detected	Virus Detected
1.	8/M	Normal	3	Productive/paroxysmal/post-tussive/Whoop	Yes	No	No	*B. pertussis*	No
2.	15/M	Normal	3	Dry/nocturnal/post-tussive	No	No	No	*B. pertussis*	No
3.	15/M	Normal	3	Dry/nocturnal/post-tussive	Yes	No	No	*B. pertussis*	No
4.	23/F	Normal	3	Dry/post-tussive	No	No	No	*B. holmesii*	No
5.	23/F	Normal	3	Dry/nocturnal	Yes	Yes	No	*B. holmesii*	hMPV
6.	9/F	Normal	4	Productive/paroxysmal	No	Yes	No	*B. holmesii*	hMPV, Rhinovirus
7.	18/M	Normal	3	Productive/nocturnal	No	Yes	No	Not Detected	Adenovirus
8.	8/M	Severe Acute Malnutrition	3	Dry/ paroxysmal	Yes	No	No	Not Detected	Adenovirus
9.	23/F	Normal	3	Productive/paroxysmal	Yes	Yes	No	Not Detected	Rhinovirus
10.	15/M	Normal	3	Productive/paroxysmal	Yes	Yes	Yes	Not Detected	hMPV
11.	15/M	Normal	3	Productive/paroxysmal	Yes	Yes	Yes	Not Detected	hMPV
12.	24/M	Normal	3	Dry/whoop	Yes	Yes	Yes	Not Detected	hMPV
13.	9/M	Normal	3	Dry/paroxysmal	No	No	Yes	Not Detected	RSV-A
14.	15/F	Normal	3	Dry/paroxysmal	Yes	Yes	No	Not Detected	RSV-A
15.	18/M	Normal	3	Productive	Yes	Yes	Yes	Not Detected	RSV A, PIV 3
16.	23/M	Normal IPD	3	Productive	No	Yes	No	Not Detected	Rhinovirus
17.	4/M	Moderate Acute Malnutrition	3	Dry/paroxysmal	Yes	Yes	Yes	Not Detected	hMPV, Adenovirus
18.	6/M	Normal	3	Productive/paroxysmal/whoop	Yes	Yes	Yes	Not Detected	RSV-A
19.	3/M	Normal	3	Productive/paroxysmal/whoop	No	No	No	Not Detected	Rhinovirus
20.	23/F	Normal	3	Productive/paroxysmal	Yes	No	No	Not Detected	RSV-A
21.	8/M	Normal	3	Dry/paroxysmal	Yes	No	Yes	Not Detected	RSV-A
22.	7/M	Normal	3	Productive/nocturnal	Yes	Yes	Yes	Not Detected	Influenza A
23.	14/F	Normal	4	Dry/paroxysmal	No	Yes	No	Not Detected	PIV-3

## Data Availability

The data presented in this study are available on request from the corresponding author.

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
