# Peer review of "Looking beyond Pertussis in Prolonged Cough Illness of Young Children"

_vaccines, 2022, doi:10.3390/vaccines10081191_

Round 1
Reviewer 1 Report
The paper by Viswanathan et al reports the aetiology of prolonged cough illness in 107 children less than two years of age, which focused on the detection of Bordetella pertussis and viral aetiology in children less than 2 years of age. The authors isolated Bordetella pertussis in three of 45 suspected cases and B. holmesii in three cases.
The paper is well written and the study is well designed.
Minor points.
1) Why the study included only patients under 2 years of age?
2) You have to include the number of Ethic Committee of your hospital.
Author Response
Comments and Suggestions for Authors
We thank the reviewers for their valuable suggestions. Please find below the pointwise response to the specific comments.
Reviewer 1
The paper by Viswanathan et al reports the aetiology of prolonged cough illness in 107 children less than two years of age, which focused on the detection of Bordetella pertussis and viral aetiology in children less than 2 years of age. The authors isolated Bordetella pertussis in three of 45 suspected cases and B. holmesii in three cases.
The paper is well written and the study is well designed.
Minor points.
- Why the study included only patients under 2 years of age?
Response: The study focused on young children as prolonged cough and incidence of pertussis is more in preschool children.
- You have to include the number of Ethic Committee of your hospital.
Response: The detail has been included in the Institutional Review Board Statement (Line 270)
Reviewer 2
Although high vaccination coverage in childhood, pertussis is still quite prevalent in many countries. The high incidence has been reported in infants, especially in those who are too young to be vaccinated. Co-infections with certain viruses have been reported, stressing the differential diagnosis is important. This study was aimed to investigate the causative agents responsible for prolonged cough illness in Indian children. The study design is proper and the finding is important. I have following concerns which should be addressed first.
Major comments:
- Title, usually infants refer to those who are younger than or equal to one year of age. Since the study is focused on those children who are unto two years of age, it is better to change the word "infants" to "young children".
Response: The point is noted and the title has been changed accordingly. The word infant has been replace by “young children” throughout the text.
- It would be important for readers to know where the study was conducted. A map to show the place is needed.
Response. The map has been provided (Fig 1)
- It is important to shortly describe the vaccination history, schedule and coverage in India.
Response: The details have now been included. (Line 51-58)
- It is important to shortly describe the reported incidence of pertussis, especially incidence of infants, in this country.
Response: The point is noted and information has been provided. (Line 58-64)
- It is important to provide some background information of this hospital where the study was done: e.g. how many outpatients per year. In this way, readers would know how the 45 patients are representative.
Response: The point is well taken and the information has now been included. (Line 74)
- It is critical to briefly describe the qPCR assays and how B. pertussis and B. holmesii are differentiated.
Response: The details have now been included. (Line 94-97;Table1)
Minor comments:
- Line 19 there should be a space between B and holmesii throughout the text.
Response: This has been done.
- Lines 183-184 should be merged.
Response: The lines have been merged. (Line 223-225)
Reviewer 2 Report
Although high vaccination coverage in childhood, pertussis is still quite prevalent in many countries. The high incidence has been reported in infants, especially in those who are too young to be vaccinated. Co-infections with certain viruses have been reported, stressing the differential diagnosis is important. This study was aimed to investigate the causative agents responsible for prolonged cough illness in Indian children. The study design is proper and the finding is important. I have following concerns which should be addressed first.
Major comments:
1. Title, usually infants refer to those who are younger than or equal to one year of age. Since the study is focused on those children who are unto two years of age, it is better to change the word "infants" to "young children".
2. It would be important for readers to know where the study was conducted. A map to show the place is needed.
3. It is important to shortly describe the vaccination history, schedule and coverage in India.
4. It is important to shortly describe the reported incidence of pertussis, especially incidence of infants, in this country.
5. It is important to provide some background information of this hospital where the study was done: e.g. how many outpatients per year. In this way, readers would know how the 45 patients are representative.
6. It is critical to briefly describe the qPCR assays and how B. pertussis and B. holmesii are differentiated.
Minor comments:
1. Line 19 there should be a space between B and holmesii throughout the text.
2. Lines 183-184 should be merged.
Author Response

(The authors gave the same response as above.)

Round 2
Reviewer 2 Report
All comments have been taken into account. The revised manuscript has been significantly improved.
Minor comments:
1. I did not find the fig 1 in this revised version.
2. Line 74, Is it true "with an outpatient attendance of 400-500 children per year? This means that there are only 1-1.5 patient per day?